# Omni-Diffusion: Unified Multimodal Understanding and Generation with Masked Discrete Diffusion

**Lijiang Li** [1 2]   **Zuwei Long** [3]   **Yunhang Shen** [3]   **Heting Gao** [3]   **Haoyu Cao** [3]
**Xing Sun** [3]   **Caifeng Shan** [1 2]   **Ran He** [4]   **Chaoyou Fu** [1 2]

## Abstract

While recent multimodal large language models (MLLMs) have made impressive strides, they mostly employ a conventional autoregressive architecture as their backbone, leaving significant room for exploring effective and efficient alternatives in architectural design. Meanwhile, recent studies have successfully applied discrete diffusion models to natural language processing, revealing their considerable potential as a promising new approach in this domain. Drawing inspiration from these pioneering studies, we introduce Omni-Diffusion, the first any-to-any multimodal language model built entirely on mask-based discrete diffusion models, which unifies understanding and generation across text, speech, and images. Omni-Diffusion employs a unified mask-based discrete diffusion model to directly capture the joint distribution over discrete multimodal tokens. This approach supports not only bimodal tasks but also more complex scenarios involving multiple modalities. On a diverse set of benchmarks, our method outperforms or performs on par with existing multimodal systems that process two or more modalities, highlighting the significant promise of diffusion models in powering the next generation of multimodal foundation models. Our codes are released at GitHub.

## 1. Introduction

In the past few years, significant advances have been made in the research on multimodal intelligence. One important direction in this field is designing a unified model that can process tasks involving data from various modalities, including text, image, speech, and beyond. To achieve this goal, many studies have developed multimodal systems by augmenting pre-trained large language models (LLMs) with multimodal perception and generation capabilities (Chen et al., 2025; Zhan et al., 2024; Fu et al., 2024; 2025b; Xu et al., 2025; Wu et al., 2024; Yang et al., 2025b), which exhibit impressive performance thanks to the strong language understanding capabilities of LLMs. However, most existing approaches in multimodal intelligence rely on autoregressive architectures, leaving substantial room to explore alternative probabilistic modeling approaches.

Recent studies have witnessed a surge of research interest in applying diffusion models to natural language processing tasks, which have emerged as a promising alternative to classical autoregressive architectures (Ye et al., 2025; Zhu et al., 2025). Diffusion models demonstrate several distinct advantages over autoregressive models (Yu et al., 2025; Ni et al., 2025). For example, the initial token sequence and the transformation direction of the diffusion generation process can be steered to control the semantic structure, output format and response style of generated content (Xin et al., 2025). Furthermore, diffusion models support parallel decoding, offering the potential for efficient generation (Wu et al., 2026b;a). Given these advantages, it is natural to explore the potential of employing diffusion models to build multimodal intelligence systems.

In this work, we introduce Omni-Diffusion, the first any-to-any multimodal language model that builds upon a mask-based discrete diffusion model for unified comprehension and generation. As shown in Figure 1, Omni-Diffusion employs a mask-based discrete diffusion model to learn the joint distribution of multimodal semantic tokens obtained by tokenizing raw text, image, and speech data. In contrast to existing multimodal models that utilize an LLM to model textual data and rely on additional output models to convert the LLM's textual hidden states into outputs of other modalities (Wu et al., 2024), this joint modeling of multimodal discrete tokens enables the model to develop an intrinsically well-aligned semantic representation space, thereby equipping it with unified comprehension and generation

---

[1]State Key Laboratory for Novel Software Technology, Nanjing University, Nanjing 210023, China [2]School of Intelligence Science and Technology, Nanjing University, Suzhou 215163, China [3]Tencent Youtu Lab [4]Institute of automation, Chinese Academy of Sciences. Correspondence to: Chaoyou Fu <bradyfu24@gmail.com>.

*Proceedings of the $43^{rd}$ International Conference on Machine Learning*, Seoul, South Korea. PMLR 306, 2026. Copyright 2026 by the author(s).

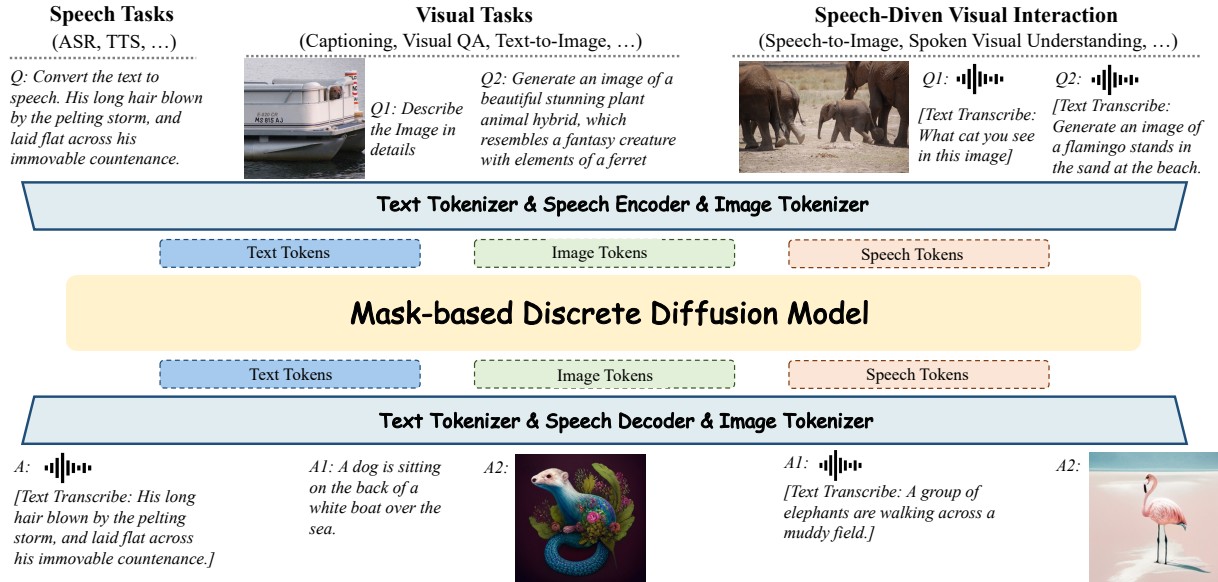

*Figure 1.* Overview of Omni-Diffusion. Our model takes multimodal tokens as input, processes them using a unified mask-based discrete diffusion model, and generates output tokens of the desired modality. By modeling the joint distribution over discrete multimodal tokens, Omni-Diffusion can handle not only bimodal tasks (e.g., automatic speech recognition (ASR), text-to-speech (TTS), Visual Question Answering (VQA), and text-to-image) but also tasks requiring the integration of more than two modalities, such as speech-to-image generation and spoken visual understanding.

capabilities across various modalities.

We introduce a suite of training and inference techniques tailored to the characteristics of mask-based discrete diffusion models, which facilitate the extension of a pre-trained diffusion language model into a multimodal system. First, we propose a three-stage progressive training pipeline to extend the model to encompass multimodal comprehension and generation capabilities. To equip Omni-Diffusion with the ability to handle any-to-any multimodal conversation, we construct a speech-driven visual interaction (SDVI) dataset that consists of samples requiring both visual and speech capabilities. Furthermore, we employ an attenuated tail-pad masking strategy to enhance the model's ability to generate responses of variable lengths. Finally, we optimize the inference process based on the characteristics of mask-based discrete diffusion models. For the image modality, we introduce a position penalty to constrain the generation order and improve the visual quality. For the speech modality, we propose a special token pre-infilling strategy that allows the model to incorporate text semantics during speech generation. Additionally, we apply an adaptive token-length initialization strategy to speech understanding and generation tasks to further boost performance.

In summary, our key contributions are as follows:

- We introduce Omni-Diffusion, the first any-to-any multimodal language model built on a mask-based discrete diffusion model. By modeling a joint distribution over multimodal discrete tokens, Omni-Diffusion enables the alignment of different modalities in a shared semantic representation space, exhibiting strong capabilities in multimodal comprehension and generation.

- We develop specialized training and inference techniques based on the characteristics of mask-based diffusion models. For training, we implement an attenuated tail-pad masking strategy to facilitate variable-length generation and a three-stage progressive training pipeline for effective multimodal alignment. For inference, we introduce a position penalty to constrain generation order and enhance visual quality, alongside a special token pre-infilling strategy to improve spoken dialogue performance.

- We conduct extensive experiments to evaluate the performance of our method. Comprehensive evaluations reveal that Omni-Diffusion achieves performance comparable to or even better than existing autoregressive multimodal systems processing two or more modalities on various benchmarks, thereby providing valuable insights into developing discrete diffusion models for multimodal intelligence.

## 2. Related Work

### 2.1. Multimodal Large Language Models

Recent multimodal research has focused on unified models for diverse input and output modalities. Several studies have

developed foundation models for multimodal comprehension. For example, OneLLM (Han et al., 2024) aligns eight modalities to an LLM using modality-specific tokenizers and progressive training. Video-SALMONN (Sun et al., 2024a) proposes to connect audio-visual encoders to an LLM via a Q-former for video and speech understanding. The VITA series (Fu et al., 2024; 2025b) introduce a duplex communication mechanism to multimodal LLMs for a natural multimodal human-computer interaction experience. Beyond multimodal comprehension, recent research extends LLMs to accommodate arbitrary input and output modalities, resulting in unified any-to-any frameworks. CoDi (Tang et al., 2023) innovatively aligns multiple different modalities within a continuous diffusion model, enabling the generation of any modality. AnyGPT (Zhan et al., 2024) processes discrete tokens across modalities with a unified LLM to enable any-to-any conversations. NExT-GPT (Wu et al., 2024) connects pretrained diffusion decoders to a frozen LLM via adapters for multimodal generation. In contrast to existing methods, we propose modeling the distribution of discrete multimodal tokens directly within a unified, mask-based discrete diffusion model.

## 2.2. Mask-based Discrete Diffusion Models

We develop our method based on Mask-based Discrete Diffusion Models (MDMs), a class of generative models that exhibit impressive performance on various tasks, such as natural language processing (Arriola et al., 2025; Ye et al., 2025; Zhu et al., 2025), image generation (Chang et al., 2023; Yang et al., 2025a), and visual understanding (You et al., 2025; Yu et al., 2025; Shi et al., 2026). MDMs model the distribution of target discrete token sequences through mask token prediction. During training, MDMs typically corrupt a clean data sequence sampled from a training dataset by replacing its tokens with a special [MASK] token. Then, a neural network is optimized to predict the original unmasked tokens given the partially masked context. In the inference process, MDMs start with a fully masked sequence and iteratively decode the mask tokens, gradually reconstructing the clean data distribution. While various pioneering works have proposed employing MDMs as the backbone of LLMs (Zhu et al., 2025; Ye et al., 2025), we further extend MDMs to a unified multimodal understanding and generation system in this work.

Discrete Flow Matching (DFM) is a class of generative models that are mathematically similar with MDMs (Gat et al., 2024; Gao et al., 2025b), and have been applied to building multimodal large language models (Wang et al., 2025a). For example, NeXT-Omni is a multimodal model based on DFM, which models a transformation from a random token sequence to the token sequences sampled from real data (Luo et al., 2026). While both DFM and MDMs model the transition from a noise distribution to a target data distribution, their decoding processes differ fundamentally. During training, DFM optimizes the probability velocity for the entire sequence, while MDMs compute the loss only on masked tokens. During inference, DFM refines the entire sequence simultaneously at every step, while MDMs iteratively decode and freeze the Top-$k$ highest-confidence tokens at each step. These different mechanisms illustrate the distinction of our model from existing DFM-based model.

## 3. Method

### 3.1. Unified Probabilistic Formulation over Multimodal Discrete Tokens

As shown in Figure 2, Omni-Diffusion is built upon a pretrained diffusion language model and performs unified learning over the joint distribution of multimodal discrete tokens. While many existing multimodal systems rely on an additional output model to project the textual features from LLMs into the generated multimodal data (Wu et al., 2024; Wang et al., 2025b), our method directly models an intrinsically unified multimodal discrete representation space, thereby achieving effective comprehension and generation of data with various modalities.

Specifically, we formulate our model as a unified mask-token predictor for discrete tokens of various modalities, including text, speech, and image. Given a data pair $(T, S, I)$ consisting of text $T$, speech $S$, and image $I$, we first tokenize them into discrete representations $\left( \{t_n\}_{n=1}^{N_t}, \{s_n\}_{n=1}^{N_s}, \{i_n\}_{n=1}^{N_i} \right)$. The token sequences of different modalities are then wrapped with special beginning and end tokens of the corresponding modality, which together form a unified token sequence $x_0 \in \mathbb{R}^L$. Following the common training process of diffusion models (Ho et al., 2020; Lou et al., 2024), we corrupt the token sequence $x_0$ by randomly replacing its tokens with a special mask token [MASK] at a ratio $r$, where $r$ is derived from the time step $t$ sampled uniformly from the interval $[0, 1]$ at each training iteration. Our model takes the corrupted token sequence at time step $t$, denoted as $x_t$, as input and predicts the clean token sequence, denoted as $\hat{x}_0 = p_\theta(x_0|x_t)$. Therefore, the training loss is the cross-entropy between model prediction $\hat{x}_0$ and the clean token sequence $x_0$:

$$L = -\mathbb{E}_{t,q(x_t|x_0)} \left[ \sum_{i=1}^{L} \mathbb{I}\left[x_t^i = [\text{MASK}]\right] \log p_\theta(x_0^i|x_t) \right]$$

(1)

where $\mathbb{I}[\cdot]$ is an indicator function that ensures the cross-entropy loss is only calculated for the masked tokens in $x_t$. With this cross-entropy loss, we train our model on text, speech, and image data in a unified mask-token prediction framework, and no modality-specific optimization is employed during training.

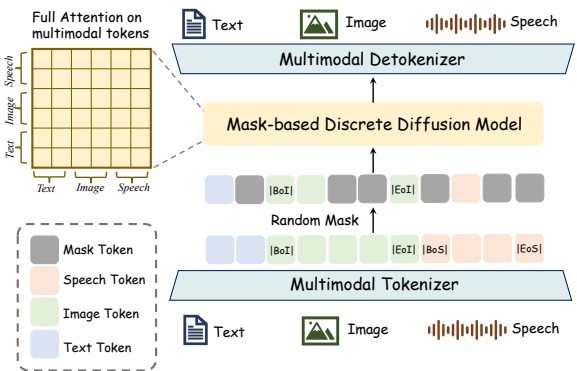

*Figure 2.* Architecture overview. Omni-Diffusion is an any-to-any multimodal system built on the mask-based discrete diffusion model. By modeling a unified distribution of multimodal discrete tokens through the mask token prediction, Omni-Diffusion enables performing comprehension and generation of various modalities, including text, image, and speech.

## 3.2. Model Architecture

Omni-Diffusion is built upon a mask-based discrete diffusion language model and is equipped with distinct tokenizers for data of various modalities. The tokenization of different modalities and the model backbone are detailed as follows.

**Image Tokenization.** We leverage the pre-trained MAGVIT-v2 (Yu et al., 2024) as an image tokenizer, following existing visual language models (Yang et al., 2025a; Xie et al., 2025). This image tokenizer compresses images into a compact representation with a downsampling factor $f = 16$ through a visual encoder. Then, a quantizer with a codebook size of 8192 is employed to convert the compact image representation into discrete tokens. We use the resulting discrete image tokens for both visual understanding and generation tasks in our implementation.

**Speech Encoder and Decoder.** We employ SenseVoiceS-mall (An et al., 2024) and the GLM-4-Voice decoder (Zeng et al., 2024) for speech encoding and decoding, respectively, following VITA-Audio (Long et al., 2025). SenseVoiceS-mall utilizes a memory-equipped self-attention network to extract semantically rich representations from input speech. These representations are then projected into the hidden dimension of our discrete diffusion model backbone via a lightweight MLP adapter. For speech generation, we leverage the GLM-4-Voice decoder. Its speech tokenizer transforms the speech into discrete tokens at a token rate of 12.5 Hz through a finite scalar quantization layer with a codebook size of 16384. Our model is trained to predict the speech tokens conditioned on multimodal input. The predicted speech tokens are finally reconstructed into waveforms by the GLM-4-Voice decoder.

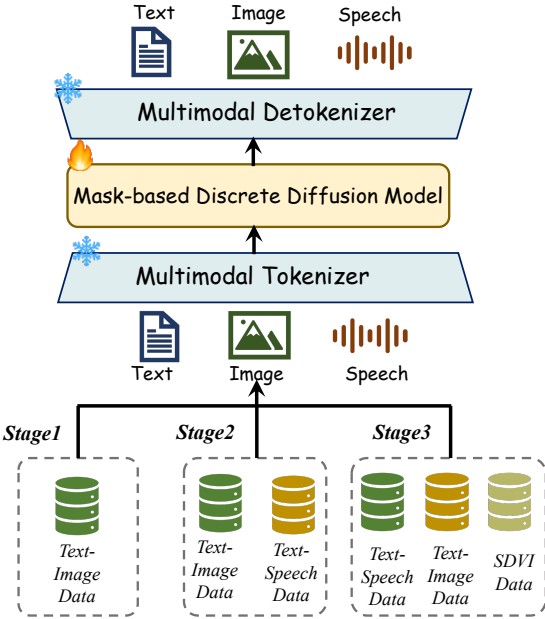

*Figure 3.* Training pipeline of Omni-Diffusion. The first stage pre-aligns the textual capability of pre-trained diffusion language model with the visual modality. The second stage further enhances the multimodal capability of diffusion model by jointly training on the speech and visual data. The last stage optimizes the model on our constructed SDVI datasets that consisting of speech-to-image and image-to-speech tasks, which further enhances the unified multimodal alignment of our model across various modality.

**LLM Backbone.** We employ Dream-7B (Ye et al., 2025), a pre-trained discrete diffusion language model, as our base model. To enable multimodal processing, we expand the vocabulary to accommodate 16384 speech tokens and 8192 image tokens. Aside from extending the vocabulary as well as the corresponding embedding and output layer, the architecture of discrete diffusion model remains unaltered.

## 3.3. Training

To achieve an efficient and stable training process, we propose a three-stage progressive training pipeline to extend the multimodal understanding and generation capabilities of the pre-trained diffusion language model effectively. Besides, we construct a Speech-Driven Visual Interaction Dataset (SDVI) consisting of spoken visual question answering and speech-to-image data to further improve the unified alignment of our model across various modalities. In addition, we propose an attenuated tail-pad masking strategy to encourage the model to generate responses of variable lengths.

**Three-Stage Progressive Training Pipeline.** As illustrated in Figure 3, our training pipeline progressively expands the set of modalities and tasks throughout the training process. This strategy ensures stability when training a unified model on data distributions with distinct characteris-

tics. Training datasets are detailed in Table 4 of Appendix. Stage 1 (Visual-Language Pre-Alignment) optimizes the pre-trained diffusion language model on text-to-image and image captioning tasks. This stage aims to align the visual modality with the semantic space of the pre-trained language model. Stage 2 (Speech–Vision–Language Joint Alignment) focuses on enhancing the alignment between text and other modalities. In this stage, we retain the visual-text datasets from Stage 1 and introduce automatic speech recognition (ASR) and text-to-speech (TTS) data to facilitate speech-text alignment. Stage 3 (Speech-Driven Visual Interaction Capability Improvement) aims to improve unified cross-modal alignment. In this stage, we fine-tune the model on our constructed Speech-Driven Visual Interaction (SDVI) dataset, which comprises samples for spoken visual question answering and speech-to-image generation that requires joint processing of speech and visual data. We also incorporate spoken question answering (SQA) and visual question answering (VQA) data during this final stage.

**SDVI Dataset Construction.** We construct the Speech-Driven Visual Interaction (SDVI) dataset that mainly includes spoken visual question answering and speech-to-image generation data to improve the model's capability for visual interaction via spoken instruction. For spoken visual question answering task, we use the LLaVA-OneVision dataset (Li et al., 2025) as the data source and employ the Cosyvoice2 (Du et al., 2024b) model to convert the textual question-answering pairs into speech. We design a processing pipeline prior to speech synthesis to ensure the dataset quality. Specifically, we first filter out all samples that contain mathematical computation or programming, as these are uncommon in daily spoken dialogue scenarios. Next, we rewrite all multiple-choice questions as open-ended question answering by replacing the answer choices with the corresponding answer words or sentences. Finally, we remove all samples with an answer length greater than 100 words, since humans usually prefer concise responses in spoken conversations. To prevent the model from overfitting to a particular human voice, we convert the question part of the processed LLaVA-OneVision dataset into speech by performing voice cloning conditioned on 1,000 randomly sampled speech samples of the GigaSpeech datasets (Chen et al., 2021) using the Cosyvoice2 (Du et al., 2024b) model, while the answer part is converted into speech using a fixed voice. The resulting dataset contains over 30,000 samples, and each sample consists of a spoken input question, an input image, a textual output answer, and the corresponding spoken output response.

For speech-to-image generation, we select Blip3o-Pretrain-JourneyDB (Chen et al., 2025) as the data source for its fluent native text and high-quality images. Similar to the spoken visual question answering task, we convert the text

into speech by performing voice cloning on 1,000 randomly speech clips sampled from the Gigaspeech dataset (Chen et al., 2021) with the Cosyvoice2 model, resulting in 30,000 speech-image pairs.

**Attenuated Tail-Pad Masking.** To facilitate variable-length generation, we adapt a tail-pad augmentation strategy that appends a random number of pad tokens to the end of each data sample, consistent with prior diffusion model training methodologies (Yu et al., 2025). During model training, both the original and pad tokens are randomly masked and serve as prediction targets. However, we observe that a simple uniform masking strategy leads to overfitting on the special pad token, resulting in the generation of excessive pad tokens during inference. To solve this issue, we propose Attenuated Tail-Pad Masking, which applies a scaling factor $\gamma$ ($\gamma < 1$) to reduce the mask ratio specifically for pad tokens. By attenuating the mask ratio, we ensure that the model's gradient updates are predominantly driven by the regular semantic tokens rather than the pad tokens, thereby mitigating overfitting and improving generation quality.

### 3.4. Inference

We utilize an entropy-based decoding strategy consistent with Dream-Instruct-7B (Ye et al., 2025). Furthermore, we propose a position penalty to enhance image generation, alongside special token pre-infilling and adaptive token length assignment to enhance spoken dialogue performance. These techniques are detailed in this section.

**Entropy-based Decoding Strategy.** During inference, we decide which tokens to decode based on the entropy of the token probabilities. To further improve generation quality, we also integrate the repetition penalty and classifier-free guidance into the inference process. Assume the token logits produced by the model backbone at time step $t$ are denoted by $z_t \in \mathbb{R}^{L \times V}$, where $L$ and $V$ represent the sequence length and vocabulary size, respectively. The logits are then adjusted by repetition penalty and classifier-free guidance, and the token probabilities are obtained by computing $p_t = \text{softmax}(z_t)$. We determine the token confidence $c_t^i$ at each position $i$ ($i = 1, 2, \cdots, L$) according to the entropy $H_t$ of the token probability $p_t^{i,v}$, which is estimated as follows:

$$c_t^i = -H_t^i = \sum_{v=1}^{V} p_t^{i,v} \cdot \log(p_t^{i,v}) \tag{2}$$

We select the top-$k$ tokens with the highest confidence and determine their values by sampling from the token probability $p_t$, while the remaining mask tokens are kept unchanged. The sampling process begins with a fully mask token sequence and iterates until all mask tokens are decoded.

**Position Penalty.** We propose a position penalty strategy to improve the image generation quality. Specifically, we observe that the model occasionally generates repetitive patterns in images. We hypothesize that these repetitive patterns arise because the model tends to decode mask tokens from the beginning and the end of the sequence towards the center, a phenomenon discussed in prior studies (Huang et al., 2025). Since diffusion models typically generate tokens with related and similar semantics within consecutive time steps, simultaneous decoding at both ends of the mask tokens sequence can result in identical patterns appearing in the top and bottom regions of the generated image. To address this problem, we propose the position penalty strategy. During the early stages of inference, we scale down the logits of the last $N^t$ tokens by a factor $\gamma_p$ ($\gamma_p < 1$). This strategy discourages the model from decoding the beginning and end of the sequence simultaneously, therefore reducing repetitive patterns and improving visual quality. It is worth noting that our position penalty differs from semi-autoregressive generation (Yang et al., 2025a), which splits the sequence into blocks and generates each block autoregressively. In contrast, our approach applies a soft constraint on the generation order without rigidly forcing the model to generate tokens in specific regions.

**Special Token Pre-Infilling.** A key advantage of discrete diffusion models is the flexibility to modify the initial mask token sequence to control the output format (Xin et al., 2025). Based on this mechanism, we propose Special Token Pre-Infilling to enhance the model performance in spoken dialogue tasks. Specifically, for an initial mask token sequence of length $L$, we replace the mask token at index $0.25L$ with a special token [begin-of-speech]. This method guides the model to generate a text response in the first $0.25L$ segment and the corresponding speech response in the remaining $0.75L$ segment simultaneously. Consequently, the model can explicitly attend to text content during speech generation, thereby improving the logic and coherence of the synthesized speech.

**Adaptive Token Length Assignment.** Similar to the special token pre-infilling strategy, we propose an adaptive assignment of the initial mask token sequence length for ASR and TTS tasks. This approach is motivated by the strong correlation between speech duration and text length, which allows us to approximate the length of one modality given the other. Accordingly, we set the initial token length to 3.5 times the text token length for the TTS task, and 0.2 times the speech token length for the ASR task. This strategy not only improves the performance of speech understanding and generation but also accelerates the sampling process by decreasing the number of tokens to be decoded.

*Table 1.* Performance on ASR and TTS tasks evaluated on the LibriSpeech (Panayotov et al., 2015) and LibriTTS (Zen et al., 2019) benchmarks. Omni-Diffusion exhibits comparable performance on ASR and superior performance on TTS compared to existing specialized speech models.

| Method | Model Type | LibriSpeech WER ↓ | LibriTTS WER ↓ |
|---|---|---|---|
| CosyVoice | TTS model | - | 2.89 |
| CosyVoice2 | TTS model | - | 2.47 |
| GLM-4-Voice | Speech LLM | 2.82 | 5.64 |
| AnyGPT | Any-to-Any | 8.5 | - |
| Omni-Diffusion | Any-to-Any | 6.69 | 2.22 |

## 4. Experiment

We evaluate the performance of Omni-Diffusion across various multimodal perception and generation benchmarks in this section. Furthermore, we assess the model's capabilities in fast sampling and image inpainting. The experimental settings and implementation details are provided in Section A of the Appendix. We also present more experimental results in Section B of the Appendix.

### 4.1. Main Results

**Speech Tasks.** We evaluate the speech capabilities of our model on ASR and TTS tasks by calculating the word error rate (WER) on the LibriSpeech (Panayotov et al., 2015) and LibriTTS (Zen et al., 2019) benchmarks. We compare our model with the existing TTS model CosyVoice (Du et al., 2024a;b), the speech LLM GLM-4-Voice (Zeng et al., 2024), and the any-to-any multimodal LLM AnyGPT (Zhan et al., 2024). To evaluate the WER for TTS, we employ Whisper-Large-V3 (Radford et al., 2023) to transcribe the generated speech into text. As shown in Table 1, compared with the autoregressive any-to-any model AnyGPT (Zhan et al., 2024), our method achieves better performance on speech tasks. In addition, Omni-Diffusion demonstrates comparable performance on the TTS task compared with the TTS expert model and shows significant improvement over the speech-specific LLM.

**Visual Tasks.** We evaluate the visual understanding and generation capabilities of Omni-Diffusion on VQA and text-to-image generation tasks, with results shown in Table 2. We compare our method with existing visual LLMs, including mPLUG-Owl (Ye et al., 2023), LLaVA (Liu et al., 2023), InstructBLIP (Dai et al., 2023), DreamLLM (Dong et al., 2024), and Emu (Sun et al., 2024c), as well as the any-to-any multimodal LLMs AnyGPT (Zhan et al., 2024) and NExT-GPT (Wu et al., 2024). For the VQA task, we evaluate model performance on several widely used benchmarks, including POPE (Li et al., 2023), MME-Perception (Fu et al., 2025a), MMMU-val (Yue et al., 2024), and Seed-2-Plus

*Table 2.* Performance on VQA and text-to-image tasks. VQA performance is evaluated on the POPE, MME (Perception), MMMU (eval) and Seed-2-Plus benchmarks, while text-to-image task is evaluated by the CLIP-T and CLIP-I scores on the MSCOCO dataset. "†" represents visual LLMs capable of understanding only. "‡" denotes models using external pretrained diffusion models. "∗" denotes evaluation results using the officially released code and model checkpoint.

| Method | Model Type | #Params | Image Question Answering | | | | Text-to-Image | |
|---|---|---|---|---|---|---|---|---|
| | | | POPE↑ | MME-P↑ | MMMU-val↑ | Seed-2-Plus↑ | CLIP-T↑ | CLIP-I↑ |
| mPLUG-Owl | Visual LLM† | 7B | - | 976.34 | - | 31.8 | - | - |
| LLaVA | Visual LLM† | 7B | 76.3 | 809.6 | - | 30.1 | - | - |
| InstructBLIP | Visual LLM† | 14B | 78.9 | 1212.8 | - | 29.2 | - | - |
| DreamLLM | Visual LLM | 7B | 69.2* | - | - | - | 0.238* | 0.697* |
| Emu‡ | Visual LLM | 14B | - | - | 30.7* | 33.5 | 0.286 | 0.656 |
| AnyGPT | Any-to-Any | 8B | 67.7* | - | - | - | - | 0.650 |
| NExT-GPT‡ | Any-to-Any | 7B | - | - | - | 26.2 | 0.225* | 0.691* |
| Omni-Diffusion (Ours) | Any-to-Any | 7B | 76.4 | 1176.1 | 31.1 | 34.7 | 0.236 | 0.662 |

(Li et al., 2024). For text-to-image generation, we evaluate CLIP-T and CLIP-I scores on 10,000 images randomly sampled from the MSCOCO 2014 validation set (Lin et al., 2014), following the methodology established by Emu (Sun et al., 2024b). CLIP-T denotes the average cosine similarity between the prompts and the generated image CLIP embeddings, while CLIP-I represents the average cosine similarity between generated and real image CLIP embeddings.

The experimental results in Table 2 demonstrate that Omni-Diffusion achieves strong performance in both visual understanding and generation. Omni-Diffusion achieves performance comparable to specialized visual LLMs in both domains. However, unlike visual LLMs designed specifically for visual tasks, our method distinguishes itself by supporting a more diverse range of modalities and tasks. Furthermore, our method achieves better visual understanding performance than existing any-to-any models. In the text-to-image task, our method achieves superior text-image alignment compared to other any-to-any models, and visual quality comparable to methods that rely on external pretrained diffusion models.

**Speech-Vision Alignment Evaluation.** We examine the model's ability to achieve unified alignment across the speech, image, and text modalities by evaluating its performance on speech-to-image generation. Specifically, we randomly sample 10,000 captions from the MSCOCO validation set and convert these captions into speech using the CosyVoice2 model. After that, we employ our model to generate images conditioned on the synthesized speech. As shown in Table 3, Omni-Diffusion achieves similar generation quality conditioned on text and speech, highlighting the model's strong alignment across various modalities.

To demonstrate the effectiveness of Omni-Diffusion for tasks involving spoken interaction with visual content, we present qualitative examples illustrating the model's ability to generate spoken responses to spoken questions regarding image content in Figure 4. From a speech perspective,

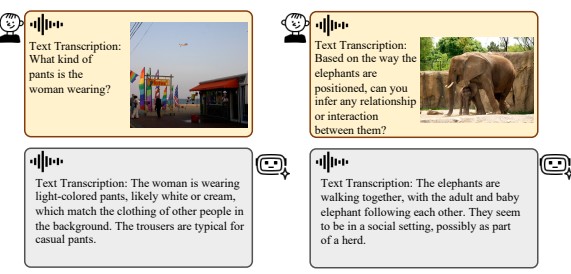

*Figure 4.* Generated samples of Omni-Diffusion on spoken interaction with visual content.

Omni-Diffusion effectively understands the user's spoken input and responds in the speech modality. Regarding visual comprehension, our model is able to capture the semantic information of the image and infer the relationships between objects. Collectively, these results illustrate the comprehensive capabilities of our model across various modalities.

### 4.2. Qualitative Results

We present qualitative examples from Omni-Diffusion for the text-to-image and speech-to-image tasks in Figure 5. More experimental results are provided in Section B.4 of the Appendix. The results demonstrate that our model is capable of generating diverse and vivid images with high-quality details. Furthermore, when conditioned on the text and speech with the same context, Omni-Diffusion is able to generate semantic consistent visual content, showcasing its strong cross-modal capabilities.

Owing to the mask-token-prediction mechanism of mask-based discrete diffusion models, our model can perform inpainting without additional fine-tuning or introducing inpainting samples into the training data. Specifically, to perform inpainting, we replace the unknown regions of the input data with [MASK] tokens and leverage our model to generate the masked part. As shown in Figure 6, Omni-Diffusion is capable of generating harmonious visual con-

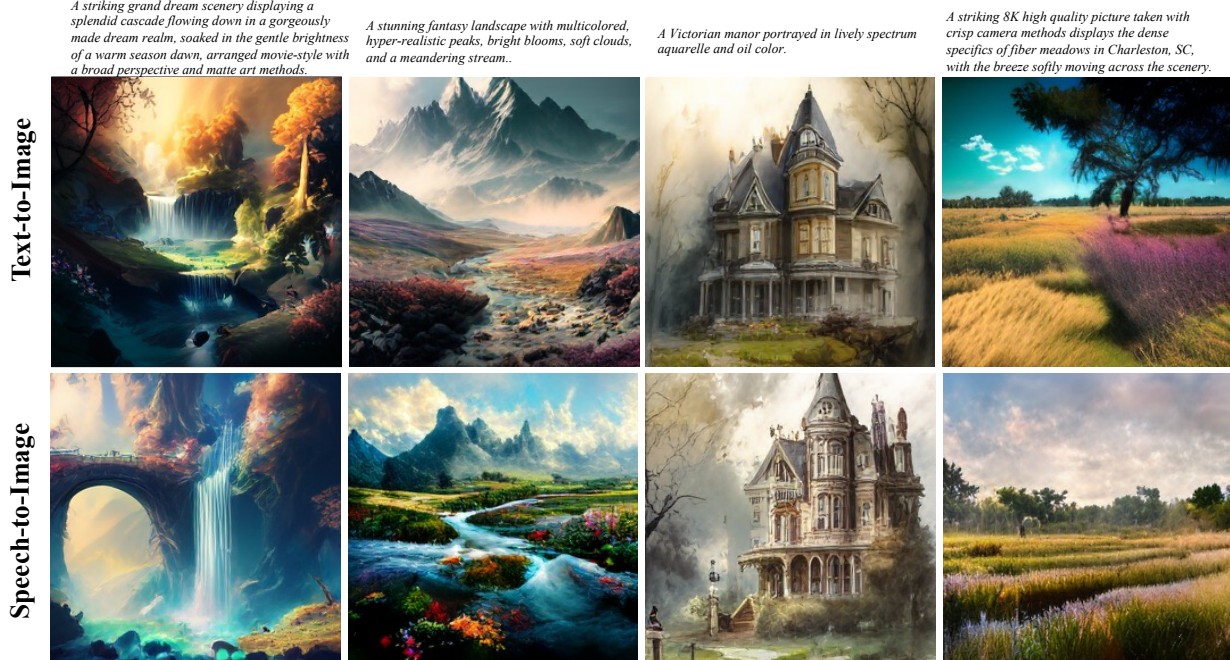

*A striking grand dream scenery displaying a splendid cascade flowing down in a gorgeously made dream realm, soaked in the gentle brightness of a warm season dawn, arranged movie-style with a broad perspective and matte art methods.*

*A stunning fantasy landscape with multicolored, hyper-realistic peaks, bright blooms, soft clouds, and a meandering stream..*

*A Victorian manor portrayed in lively spectrum aquarelle and oil color.*

*A striking 8K high quality picture taken with crisp camera methods displays the dense specifics of fiber meadows in Charleston, SC, with the breeze softly moving across the scenery.*

**Text-to-Image**

**Speech-to-Image**

*Figure 5.* Generated samples of Omni-Diffusion on text-to-image and speech-to-image tasks.

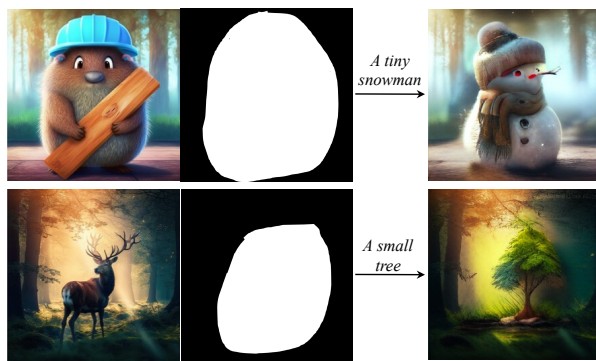

*A tiny snowman*

*A small tree*

*Figure 6.* Generated samples of Omni-Diffusion on inpainting task.

tent conditioned on the unmasked part of the input image and the prompt. These results demonstrate the advantages of diffusion-based generation systems compared with autoregressive models for downstream visual generation tasks.

### 4.3. Sampling Efficiency

Sampling efficiency is a key advantage of discrete diffusion models over autoregressive architectures. Unlike classical autoregressive models that generate tokens sequentially, discrete diffusion models can generate multiple tokens in a single forward pass via parallel decoding. In these experiments, we evaluate the sampling efficiency of Omni-Diffusion on text-to-image and TTS tasks. Specifically, for text-to-image generation, we initialize the generation process with 256

*Table 3.* Performance of image generation and TTS across various numbers of inference time steps. The Latency of Text-to-Image and Speech-to-Image tasks is the average seconds of model to generate one image, while the latency of TTS task is the average RTF estimated as $\frac{\text{Inference Latency}}{\text{Speech Duration}}$. Metrics: CLIP-T/CLIP-I($\uparrow$) for image generation and WER ($\downarrow$) for TTS. $L$ denotes the sequence length of TTS task. Our model maintains strong performance even as the number of time steps decreases.

| Task | Steps | Latency ↓ | Metrics* |
|---|---|---|---|
| Text-to-Image | 256 | 28.57 | 0.2360 / 0.6614 |
| | 50 | 5.52 | 0.2354 / 0.6593 |
| | 10 | 1.29 | 0.2325 / 0.6524 |
| Speech-to-Image | 256 | 39.90 | 0.2322 / 0.6450 |
| | 50 | 7.74 | 0.2323 / 0.6454 |
| | 10 | 4.25 | 0.2289 / 0.6376 |
| TTS | 0.5L | 0.427 | 2.22 |
| | 0.25L | 0.358 | 2.67 |
| | 0.125L | 0.341 | 5.54 |

[MASK] tokens and evaluate the CLIP score under various numbers of time steps. For TTS, we employ adaptive token length assignment to determine the sequence length and set the number of inference steps as a ratio of the total [MASK] tokens. We evaluate the TTS performance using WER metric on the LibriTTS benchmark. As shown in Table 3, our model maintains strong generation quality on text-to-image generation when the number of time steps is reduced from 50 to as few as 10. Similarly, for TTS task, Omni-Diffusion maintains consistent performance when the number of time

Steps 10   Steps 50   Steps 128   Steps 256

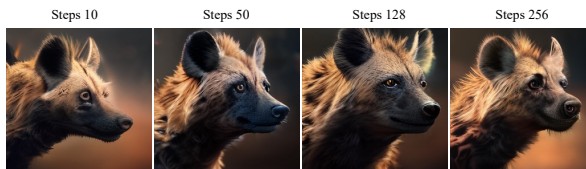

*Figure 7.* Generated samples of Omni-Diffusion under various number of time steps, the input prompt is: *This is an 8k HD depiction of a hyena.*

steps exceeds $0.25$ times the total number of [MASK] tokens. We also estimate the inference latency of our model in Table 3. The latency of text-to-image generation is evaluated as the average seconds spent by Omni-Diffusion to generate one image, while the latency of TTS is estimate as the average RTF caculated by $\text{RTF} = \frac{\text{Inference Latency}}{\text{Speech Duration}}$. The results in Table 3 demonstrate the inference latency decline significantly when reducing the number of inference steps. Additionally, Figure 7 visualizes the images generated by Omni-Diffusion under various number of time steps. Our method is able to captures the semantic content of the prompt even with extremely few steps, while the visual quality and fine-grained details improve significantly as the number of steps increases. These results highlight the potential of mask-based discrete diffusion models on efficient multimodal comprehension and generation.

## 5. Conclusion

In this work, we present Omni-Diffusion, an any-to-any multimodal language model built purely on mask-based discrete diffusion models. By modeling a joint distribution over multimodal tokens, Omni-Diffusion performs unified comprehension and generation across various modalities, including text, image, and speech. Extensive experiments demonstrate that our method achieves performance comparable to or even better than existing AR-based methods. Overall, our research demonstrates the significant potential of diffusion models to serve as the foundation models for multimodal AI systems.

## Impact Statement

This paper presents work whose goal is to advance the field of machine learning. There are many potential societal consequences of our work, none of which we feel must be specifically highlighted here.

## Acknowledgments

This work is funded by Fundamental and Interdisciplinary Disciplines Breakthrough Plan of the Ministry of Education of China (JYB2025XDXM902), National Natural Science Foundation of China (Grant No. 62506158 and No. 62441234), Basic Research Program of Jiangsu (BK20251183), and CCF-Tencent Rhino-Bird Open Research Fund.

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

# A. Implementation Details

Our model is initialized with the weights of the pre-trained Dream-7B-Instruct (Ye et al., 2025) discrete diffusion language model. For modality-specific processing, we incorporate MAGViT-v2 (Yu et al., 2024) for image tokenization, SenseVoiceSmall (An et al., 2024) for speech encoding, and GLM-4-Voice decoder (Zeng et al., 2024) for speech decoding. The training datasets of Omni-Diffusion in the three-stage progressive training pipeline are detailed in Table 4. Optimization is performed using AdamW with hyperparameters set to $\beta_1 = 0.9$, $\beta_2 = 0.95$, and $\epsilon = 1 \times 10^{-8}$. We use a learning rate of $1 \times 10^{-4}$ at stage 1 and stage 2, while the learning rate is reduced to $1 \times 10^{-5}$ at stage 3. The batch size and maximum sequence length are set to 128 and 3072 tokens across all training stages, respectively. We set $\gamma = 0.6$ for attenuated tail-pad masking. The parameter $N^T$ for the position penalty is set to $L - 100$ ($L$ denotes the token sequence length), and $\gamma_p$ is designed to decay as the token index increases, with a minimum value of 0.5.

*Table 4.* Summary of datasets used in Omni-Diffusion.

| Modality | Task | Name | Total Number | Training Stages |
|---|---|---|---|---|
| Pure Text | Text QA | Tulu 3 SFT mixture (Lambert et al., 2025) | 670K | 1,2,3 |
| Text-Image | Image Caption | Laion-2B (Schuhmann et al., 2022) | 10M | 1,2 |
| | Visual QA | LLaVA-OneVision (Li et al., 2025)
In-house Dataset | 820K
2000K | 2,3 |
| | Text-to-Image | JourneyDB (Sun et al., 2023)
JourneyDB (Sun et al., 2023) | 4000K
4000K | 1,2,3 |
| Text-Speech | ASR | Librispeech (Panayotov et al., 2015)
Common Voice 17 (Panayotov et al., 2015)
GigaSpeech (Chen et al., 2021)
People's Speech (Galvez et al., 2021)
VoxPopuli (Wang et al., 2021) | 100 Hours
100 Hours
1,000 Hours
100 Hours
54 Hours | 2,3 |
| | TTS | LibriTTS (Zen et al., 2019)
GLOBE (Wang et al., 2024)
Emilia (He et al., 2024) | 58 Hours
50 Hours
5,000 Hours | 2,3 |
| | Speech QA | VoiceAssistant-400K (Xie & Wu, 2024)
AudioQA-1.0M (Gao et al., 2025a) | 250K
180K | 2,3 |
| Text-Image-Speech | Spoken Visual QA
Speech-to-Image | SDVI
SDVI | 30K
30K | 3 |

# B. Additional Experiments

## B.1. Quantitative evaluation on Spoken VQA

We assess the model capability of spoken interaction with visual content by conducting a quantitative evaluation on the Spoken VQA task. Since there is no benchmark for Spoken VQA, we converted MME text questions into speech for evaluation through the CosyVoice2 model. The experimental result in Table 5 shows that Omni-Diffusion maintains strong spoken visual interaction capability.

## B.2. Ablation Study

We conduct ablation studies on several key designs of our method.

**Effect of position penalty.** We introduce position penalty to prevent the model from generating repetitive pattern and improve the generation quality of image, as detailed in Section 3.4. Table 6 demonstrates that the visual quality declines without the position penalty strategy, which confirms its effectiveness in enhancing visual generation quality.

**Effect of special token pre-infilling.** We propose special token pre-infilling strategy to improve the generation quality on spoken interaction with visual content. The qualitative results on Figure 8 shows that the model tends to generate overly brief response without special token pre-infilling strategy.

*Table 5.* Performance on Spoken VQA task. We test Omni-Diffusion by transferring the text questions of MME (Perception) benchmark into speech, and estimate the model performance on both text output and speech output.

| Modality | MME-P |
|---|---|
| Text + Image → Text | 1176.1 |
| Speech + Image → Text | 997.1 |
| Speech + Image → Speech | 967.1 |

*Table 6.* Model performance of Omni-Diffusion on text-to-image task with and without the position penalty strategy. The evaluation is conducted on the COCO benchmark. The experimental results demonstrate that the position penalty strategy can improve the visual generation quality of our model.

| Method | CLIP-T ↑ | CLIP-I ↑ |
|---|---|---|
| *w/o* Position Penalty | 0.222 | 0.647 |
| *w/* Position Penalty | 0.236 | 0.662 |

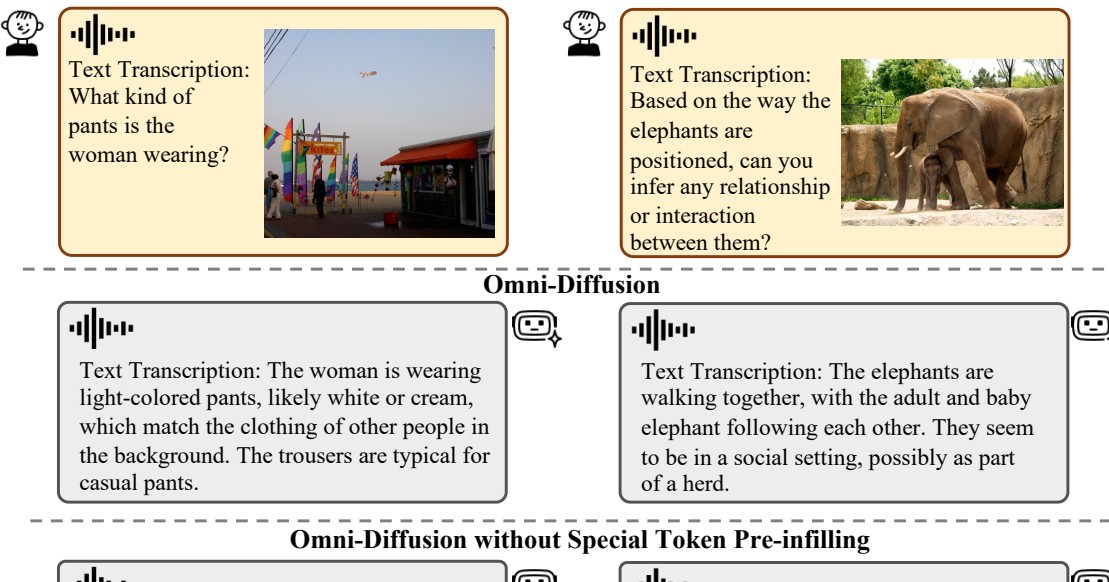

*Figure 8.* Generated samples of Omni-Diffusion on spoken interaction with visual content with and without Special Token Pre-infilling strategy.

### B.3. Latency comparison with AR baselines

We compare the inference latency of our method against AR models across different tasks in Table 7 and Table 8. The latency of text-to-image generation is assessed by the average seconds of generating one image in the MSCOCO benchmark, while the latency of the TTS task is estimated by the average RTF calculated as $RTF = \frac{\text{Inference Latency}}{\text{Speech Duration}}$. The model performance and latency of TTS task is evaluated on the LibriTTS benchmark. When reducing the number of inference steps, our model demonstrates better generation efficiency compared with AR baselines.

### B.4. More Examples of Image Generation

We present more examples of generated image conditioned on both text and speech in Figure 9 and Figure 10.

## C. Limitation and Future Direction

Although Omni-Diffusion has the capabilities of perception and generation across various modalities, the model can be further improved by extending to more extensive downstream tasks, such as instruction-based visual content editing. We plan to explore this direction through scaling data and model parameters in the future.

*Table 7.* Average latency (s) for generating an image on MSCOCO benchmark of our model compared with other baselines.

| Method | Latency | CLIP-T ↑ | CLIP-I ↑ |
|---|---|---|---|
| Emu | 3.02 | 0.286 | 0.656 |
| NeXT-GPT | 6.23 | 0.225 | 0.691 |
| Omni-Diffusion (256 steps) | 28.5 | 0.236 | 0.661 |
| Omni-Diffusion (10 steps) | 1.29 | 0.232 | 0.652 |

*Table 8.* Latency Comparison on TTS task. $L$ denotes the token sequence length determined by the Adaptive Token Length Assignment strategy.

| Method | RTF | WER ↓ |
|---|---|---|
| GLM-4-Voice | 0.685 | 5.64 |
| Omni-Diffusion $0.5L$ | 0.427 | 2.22 |
| Omni-Diffusion $0.125L$ | 0.341 | 5.54 |

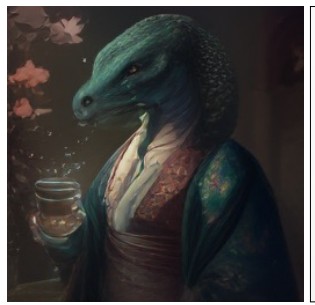 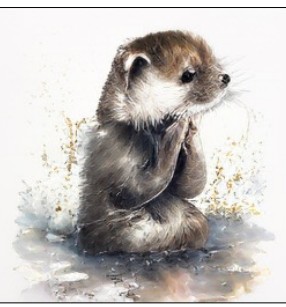 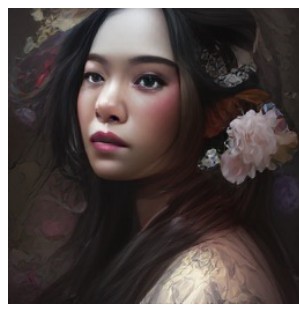 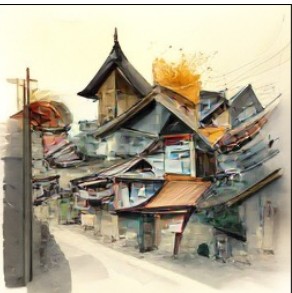

*A sweet and calm drake in a kimono is quietly sipping brew with its orbs shut.*

*A watercolor painting of baby otter with its tiny paws clasped together that is peacefully praying..*

*A dreamlike Asian female with blooms in her locks taken in an expert picture with movie-style illumination, displaying theatrical brightness, soft shades, and extreme specifics.*

*A watercolor artwork illustrating a housing district in Japan.*

*Figure 9.* Generated samples of Omni-Diffusion on text-to-image task.

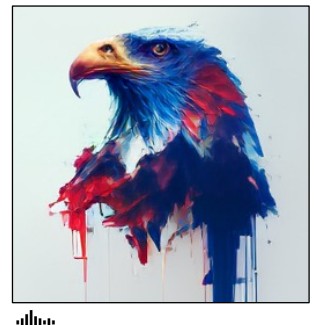 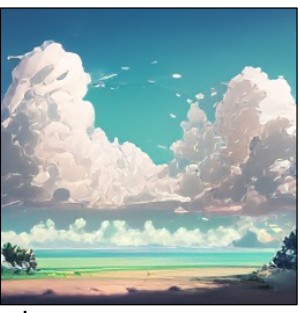 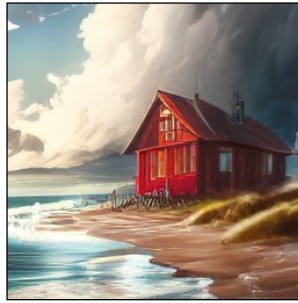 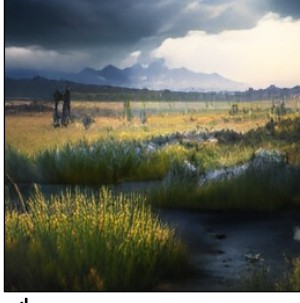

*Text Transcription: An image of America's mountains mixed with a grand royal male eagle in double exposure, with dripping red and blue colors.*

*Text Transcription: A background cloud view with an anime style.*

*Text Transcription: A red wooden cabin sits on a coast, surrounded by waves crashing onto the shore, with the huge ocean stretching into the distance under a sky full of realistic clouds.*

*Text Transcription: A photorealistic picture of a Yellowstone grass and sagebrush meadow with a pine forest in the distance, surrounded by big mountains. Rays of sunshine break through the sky, creating a warm glow on the lively landscape. Distant rainclouds suggest the possibility of rainfall in the peaceful scene.*

*Figure 10.* Generated samples of Omni-Diffusion on speech-to-image task.

