# OpenReview forum: "Omni-Diffusion: Unified Multimodal Understanding and Generation with Masked Discrete Diffusion"
_ICML.cc/2026/Conference — ICML 2026 regular_

### Official Review · Reviewer_wiWY · 2026-03-10

**Soundness:** 3
**Presentation:** 2
**Significance:** 3
**Originality:** 2
**Overall Recommendation:** 4
**Confidence:** 4

**Summary:**

This paper introduces Any-Diffusion, an any-to-any multimodal language model built entirely on mask-based discrete diffusion models to unify comprehension and generation across text, speech, and images. Overall, the authors analyze a notable topic by exploring discrete diffusion as a promising alternative to conventional autoregressive architectures in the realm of multimodal foundation models. The article's notable contribution consists of the formulation of a unified joint distribution over discrete multimodal tokens and the introduction of several customized training and inference techniques (such as attenuated tail-pad masking and position penalty).

**Compliance With Llm Reviewing Policy:**

Affirmed.

**Ethical Review Concerns:**

Missing Section: The submission lacks a dedicated Impact Statement. In accordance with ICML 2026 guidelines, the authors must include a section discussing the broader societal implications.

**Final Justification:**

The responses have addressed several of my main concerns. That said, I still think a few limitations remain insufficiently resolved or discussed:

The response on training time consumption vs. AR models is still incomplete.

The current evidence still covers a relatively limited set of benchmarks and downstream scenarios.

The authors should provide the complete ablation analysis.

**Key Questions For Authors:**

1. Under this specific training paradigm, is the model only capable of supporting $X_s \rightarrow X_s$ generation, and not $X_s \rightarrow X$ generation? (Where $X_s$ refers to a combination of multiple modalities, such as Image+Audio+Text, and $X$ refers to a single modality).

2. Furthermore, can the Mask-based Discrete Diffusion Model used for recovering masked tokens guarantee that the complete inputs are not modified during the process? For example, if the input consists of a complete Image and Audio alongside masked Text, can the model guarantee that the input Image and Audio remain strictly unchanged in the final output?

3. When comparing the training resource consumption and training time with AR, what is the difference?.

**Limitations:**

Authors should provide more diverse evaluation scenarios to demonstrate the robustness of the model, discuss the potential or advantages of discrete diffusion models compared to AR; the computational consumption and inference time should also be compared.

**Strengths And Weaknesses:**

**Strength:**
1. The paper explores the potential of discrete diffusion models in the realm of multimodal foundation models (Any-to-Any generation), which is a novel and interesting topic and should be encouraged.
2. This paper introduces a Speech Driven Visual Interaction (SDVI) dataset, which is a valuable resource for the community of cross-modal understanding and generation.
3. Some experiments basically validate the feasibility of discrete diffusion models in the realm of multimodal foundation models (Any-to-Any generation).


**Weaknesses:**

1. **Mismatch between the "Any-to-Any" claims and experimental validation:** Although targeting the highly general concept of "Any-to-Any" generation via diffusion models is an innovative point, the paper fails to provide matching qualitative and quantitative performance demonstrations.

   * **Missing Baseline:** CoDi [1] is a highly representative open-source work in Any-to-Any generation. It should definitely be included as a baseline in the main experiments for comparison.

   * **Missing Benchmark Results:** The paper only quantitatively evaluates Any-Diffusion on ASR, TTS, VQA, and Text-to-Image (TTI) tasks. This is insufficient to demonstrate true "Any-to-Any" capabilities. The authors are advised to provide the performance comparisons on a broader range of downstream task benchmarks, referencing the evaluation protocols in works like NExT-GPT [2].

   * **Lack of Diverse Downstream Task Demonstrations:** As claimed, Any-Diffusion should support complex joint generation scenarios (e.g., Image+Text+Audio $\rightarrow$ Image+Text+Audio). This should naturally extend to numerous downstream tasks such as image editing or voice style transfer. However, the paper lacks both qualitative and quantitative demonstrations for these tasks, making it easy to question whether the model truly achieves robust Any-to-Any generation.
   * **Lack of experiments for Spoken VQA:** Spoken VQA task from the self-constructed SDVI is not included in the experiments and I wonder whether the answer is presented in text modality or speech modality, the performance is different.

2. **Unclear Necessity of the Proposed Architecture:** Although the authors successfully introduce discrete diffusion models (dLLMs) into an Any-to-Any generation framework with some success, the paper does not clearly explain the fundamental *necessity* of this integration. Instead, the evaluation format is largely confined to the standard presentation style of previous autoregressive (AR) Any-to-Any models. The authors must further clarify why this diffusion-based integration is necessary and explicitly discuss/demonstrate its definitive advantages over autoregressive models.

3. **Lack of Ablation Studies:** The paper proposes several strategies to optimize model training and inference, namely Attenuated Tail-Pad Masking, Position Penalty, Special Token Pre-Infilling, and Adaptive Token Length Assignment. However, there are no ablation studies provided to substantiate the effectiveness and individual contributions of these techniques.

4. **Insufficient Evaluation Metrics for TTS:** In the Text-to-Speech (TTS) task, the paper relies solely on Word Error Rate (WER) as an evaluation metric. However, speech generation tasks typically require evaluating Speaker Similarity (SIM) as well as subjective quality via Mean Opinion Score (MOS). Relying solely on WER cannot prove that the generated speech possesses high-quality naturalness or emotional expressiveness.

[1] Any-to-any generation via composable diffusion

[2] NExT-GPT: Any-to-Any Multimodal LLM

---

> ### Author Rebuttal · Authors · 2026-03-31
>
> We thank the reviewer for the feedback and provide point-by-point responses below. The discussion will be included in the final version.
>
> **Q1.1: Missing Baseline**
>
> **R1.1:** We appreciate the comment. We agree that CoDi is a representative work and will add a discussion on it in our related work as follows: “**CoDi innovatively aligns multiple different modalities within a continuous diffusion model, enabling the generation of any modality.**”
>
> We also include CoDi as a baseline for text-to-image. CoDi achieves CLIP-I / T scores of **0.694/0.247**, slightly higher than ours (**0.662/0.236**). However, we clarify that **the two models operate in fundamentally different paradigms**: 1) **Architecture:** CoDi uses aligned continuous diffusion, while we explore discrete diffusion LLM. **2) Capability:** CoDi focuses primarily on generation, while Any-Diffusion inherits LLMs' language capabilities for reasoning (e.g., VQA, spoken dialogues).
>
> **Q1.2: Missing Benchmark**
>
> **R1.2:** We appreciate this suggestion. We have evaluated most representative benchmarks for **understanding (VQA, ASR)** and **generation (TTI, TTS, Speech-to-Image)**, alongside qualitative evaluation on downstream tasks like **inpainting**. These evaluations demonstrate **our model’s any-to-any capabilities within its supported modalities**. While evaluating more tasks is valuable, we respectfully clarify that our model uses only **75.18B** training tokens (**see the R3 to Reviewer ukSj for details**), significantly fewer than AR baselines (e.g., 176.4B for AnyGPT). **Achieving such capabilities under extremely limited resources highlights our method's effectiveness**. We will scale data for broader tasks in future work.
>
> **Q1.3 & Q1.4: Lack of evaluation for Downstream Task and Spoken VQA**
>
> **R1.3 & R1.4:** We clarify that our model does support diverse tasks, such as **Image Editing** (Figure 6) and **Spoken VQA** (Figure 4). We conducted **a quantitative study on Spoken VQA** by converting MME text questions into speech. Our MME-P scores for VQA, Spoken VQA (Speech+Image→Text), and Spoken VQA (Speech+Image→Speech) are **1176.1, 997.1, and 967.1**. These results confirm **a well-aligned multimodal semantic space and strong performance** of our model.
>
> **Q2. Unclear Necessity of the Proposed Architecture**
>
> **R2:** We respectfully clarify that Any-Diffusion has unique advantages over AR models in these aspects: 1) **Parallel Generation.** Our model enables **low-latency generation** by decoding multiple tokens per forward pass. On text-to-image task (10 steps), our model achieves **1.29s** latency with a 0.652 CLIP-I, significantly faster than Emu (**3.22s**) and NeXT-GPT (**6.23s**). 2) **Mask-Prediction.** This formulation allows our model to perform tasks like **inpainting**, which is non-trivial for standard AR models.
>
> In summary, our work validates **discrete diffusion's strong potential as a unified multimodal system**.
>
> **Q3. Lack of Ablation Studies**
>
> **R3:** We agree that ablation would be valuable. However, it is **challenging to complete during the short rebuttal.** We provide key ablations on several designs: **1) Position Penalty:** Removing it reduces text-to-image CLIP-I/T from **0.236/0.662** to **0.222/0.647**, confirming its benefit. **2) Special Token Pre-infilling:** Without it, spoken VQA responses become overly brief. For example, the first case in Figure 4 becomes “*The woman is wearing jeans*” without this strategy.
>
> **Q4. Insufficient Evaluation Metrics for TTS**
>
> **R4:** We evaluate the MOS on LibriTTS following this valuable comment. Our model achieves a WER of **2.22** and a MOS of **3.84**, compared to **5.64** and **3.95** for the Speech LLM GLM-4-Voice. These results show our method achieves a **comparable MOS score and better WER**. While SIM is important for specialized Speech LLMs, our primary focus is **comprehensive multimodal capabilities** rather than specific tasks like voice cloning.
>
> **Q5. Can the model support $X_s$ → $X$  generation?**
>
> **R5: Yes**. It supports Spoken VQA (image + speech → speech) and inpainting (text + image → image). We will scale data for more diverse tasks in the future.
>
> **Q6. Can the model guarantee that complete inputs remain unchanged when recovering masked tokens?**
>
> **R6: Yes.** It only predicts masked tokens conditioned on unmasked ones. Thus, input tokens remain strictly unchanged in the final output.
>
> **Q7. Training resource comparison with AR models.**
>
> **R7:** Our model uses fewer training tokens (**75.18B**) than AR models like GLM-4-Voice (1055B) and Emu (150B). Furthermore, since both paradigms use Transformer backbones, the GPU memory is identical for the same model size and sequence length.
>
> **Q8. Missing limitation and impact statement.**
>
> **R8:** We appreciate this comment. We detail the **unique advantages** of our model in **R2**, and **computational costs** in **R7 and R2**. We will make sure to add the limitation and impact statement into our paper.

---

> > ### Author Rebuttal · Reviewer_wiWY · 2026-04-04
> >
> > Thanks for the response and additional experimental results. It addressed several of my main concerns, and I will increase score to 4. That said, I still think a few limitations remain insufficiently resolved or discussed:
> >
> > 1. The response on training time consumption vs. AR models is still incomplete.
> >
> > 2. The current evidence still covers a relatively limited set of benchmarks and downstream scenarios.
> >
> > 3. The authors should provide the complete ablation analysis.

---

> > > ### Author Response · Authors · 2026-04-08
> > >
> > > We sincerely thank you for increasing the score and the thoughtful feedback. Below, we provide the detailed responses to the new concerns. We are making every effort to improve our paper according to your constructive comments.
> > >
> > > **Q1: The response on training time consumption vs. AR models is still incomplete.**
> > >
> > > **R1**: Thanks for the constructive comment. We would like to clarify that directly comparing training time with AR baselines is highly challenging due to **variances in computational devices and training settings**. For instance, DreamLLM was trained on A800 GPUs, while LLaVA, Emu and InstructBLIP were optimized on A100 GPUs. In addition, the any-to-any models with similar functions to ours, AnyGPT and NeXT-GPT, do not report their exact training times. More importantly, current mainstream discrete diffusion models (e.g., Dream and LLaDA) lack computational optimizations like FlashAttention, which makes them significantly different from AR baselines.
> > >
> > > In summary, we agree that the training resource comparison is meaningful. **We have completed the comparison of training tokens**. Our model consumes 75.18B tokens, which is fewer than other baselines, such as Emu (150B) and Any-GPT (176.4B). **To address the concern on training time, we are currently conducting controlled experiments** by aligning the training devices, data, and settings with AR baselines to compare the training time fairly. We plan to include such studies in the final version of the paper.
> > >
> > > **Q2: The current evidence still covers a relatively limited set of benchmarks and downstream scenarios.**
> > >
> > > **R2**: We thank you for this valuable comment. We fully agree that evaluation on more benchmarks would be meaningful, and are currently searching for appropriate benchmarks to evaluate our model, as there exists a broader range of any-to-any tasks. **We will include the results of additional benchmarks and downstream tasks in our final version**. Currently, we have evaluated our model across a wide range of representative benchmarks covering both **multimodal understanding and generation**. This includes ASR (LibriSpeech), VQA (MME, POPE, Seed-2-Plus), Spoken VQA (MME), Text-to-Image (COCO), TTS (LibriTTS), and Speech-to-Image (COCO). Additionally, **we conducted human studies on Text-to-Image and Speech-to-Image generation**, scoring Prompt Alignment (PA) and Aesthetic Quality (AQ) on a scale of 1 to 5. The detailed comparisons with baselines are summarized below:
> > >
> > > **Text-Speech Task:**
> > >
> > > | Model | LibriSpeech WER↓ | LibriTTS WER↓ / MOS $\uparrow$ |
> > > | --- | --- | --- |
> > > | GLM-4-Voice | 2.82 | 5.64 / 3.95 |
> > > | Ours | 6.69 | 2.22 / 3.84 |
> > >
> > > **VQA Task:**
> > >
> > > | Model | POPE $\uparrow$ | MME-P $\uparrow$ | Seed-2-Plus $\uparrow$ |
> > > | --- | --- | --- | --- |
> > > | LLaVA | 76.3 | 809.6 | - |
> > > | Emu | - | - | 33.5 |
> > > | NeXT-GPT | - | - | 26.2 |
> > > | Ours | 76.4 | 1176.1 | 34.7 |
> > >
> > > **Text-to-Image Task:**
> > >
> > > | Model | CLIP-T $\uparrow$ | CLIP-I $\uparrow$ | PA $\uparrow$ | AQ $\uparrow$ |
> > > | --- | --- | --- | --- | --- |
> > > | Emu | 0.286 | 0.656 | 3.66 | 3.97 |
> > > | NeXT-GPT | 0.225 | 0.691 | 3.63 | 3.96 |
> > > | Ours | 0.236 | 0.662 | 3.83 | 3.89 |
> > >
> > > **Speech-Image Task:**
> > >
> > > | Model | Speech-to-Image (CLIP-T / CLIP-I / PA / AQ) ↑ | Spoken MME-P ↑ |
> > > | --- | --- | --- |
> > > | Ours | 0.2322 / 0.6450 / 3.89 / 3.91 | 967.1 |
> > >
> > > **Q3: The authors should provide the complete ablation analysis.**
> > >
> > > **R3**: Thanks for the comment. We have already conducted the ablations for several key designs, including:
> > >
> > > - **Position Penalty:** Removing the position penalty in the text-to-image task causes the CLIP-T / I scores to drop from **0.236 / 0.662** to **0.222 / 0.647**, confirming its effectiveness in enhancing visual generation quality.
> > > - **Special Token Pre-infilling:** We conducted qualitative ablations in the spoken VQA task based on **the first example in Figure 4**. Without special token pre-infilling, the model tends to generate overly brief responses (e.g., degrading to merely "*The woman is wearing jeans*"). We will provide more cases in the revised paper.
> > >
> > > We agree that a full ablation study would strengthen our paper. However, it is very difficult to complete this within the short rebuttal period. We are currently conducting the remaining ablations on our training strategies (e.g., the training pipeline and attenuated tail-pad masking), and plan to include them in the final version.

---

### Official Review · Reviewer_qyGk · 2026-03-12

**Soundness:** 3
**Presentation:** 3
**Significance:** 3
**Originality:** 3
**Overall Recommendation:** 4
**Confidence:** 4

**Summary:**

The paper introduces Any-Diffusion, the first "any-to-any" multimodal language model entirely based on mask-based discrete diffusion models. The authors break away from the current paradigm where multimodal large models generally rely on autoregressive architectures. By jointly modeling the distribution of discrete multimodal tokens for text, speech, and images within a unified framework, they achieve cross-modal understanding and generation.

**Compliance With Llm Reviewing Policy:**

Affirmed.

**Final Justification:**

I thank the authors for their clarifications. All of my concerns have been addressed, and I will maintain my positive score.

**Key Questions For Authors:**

Compared to existing autoregressive multi-modal large models, what specific advantages and disadvantages does Any-Diffusion have in terms of computational cost (e.g., training memory usage, KV Cache equivalent mechanism during inference)?

**Limitations:**

yes

**Strengths And Weaknesses:**

**Strengths**:
1. In the context of high homogenization of multimodal foundational models (primarily autoregressive paradigms), utilizing discrete diffusion models to unify multimodal tasks represents a forward-thinking and highly novel exploration direction;
2. The paper proposes a series of training and inference techniques specifically tailored for the masked discrete diffusion mechanism. Among these, to address the common issue of repetitive patterns in image generation, the authors introduce the Position Penalty strategy, which effectively enhances visual generation quality by applying soft constraints to the beginning and end of sequences.


**Weaknesses**:
1. In the Adaptive Token Length Assignment strategy, the authors directly fix the initial token lengths of TTS and ASR to 3.5 times the text length and 0.2 times the speech token length, respectively. This heuristic rule, based on strong assumptions, may prove to be very fragile when faced with real-world complex scenarios involving extremely fast or slow speech rates and significant differences between long and short sentences.
2. The model simultaneously accommodates 16,384 speech tokens and 8,192 image tokens by simply expanding the vocabulary. Although this unifies the cross-entropy loss function, the paper lacks in-depth ablation experiments to demonstrate whether there is gradient conflict or modality competition among these three vastly different modalities within the same parameter space.

---

> ### Author Rebuttal · Authors · 2026-03-31
>
> We are grateful for the reviewer's time and constructive feedback.  Please find our point-by-point responses below. We will include the discussion and additional results in our final version.
>
> **Q1: The Adaptive Token Length Assignment strategy may be fragile in complex real-world scenarios.**
>
> **R1:** We appreciate the thoughtful comment. We agree that the adaptive token length assignment strategy may face challenges in extreme scenarios. However, we clarify that **this strategy is currently indispensable**. Specifically, in ASR and TTS tasks, the input lengths vary drastically. Forcing a fixed token length causes the model to generate excessive padding tokens for short inputs or truncate information for long inputs, which degrades generation quality. Existing works on discrete diffusion models have also pointed out this issue [1]. Therefore, adaptive length assignment is an essential architectural adaptation for ASR/TTS tasks. Through empirical evaluation, we find that the hyper-parameters 3.5 and 0.2 perform best on ASR/TTS. We acknowledge the limitation of this strategy in real-world scenarios, and will explore more flexible, dynamic length prediction strategies in future work.
>
> [1]. Dimple: Discrete diffusion multimodal large language model with parallel decoding.
>
> **Q2. Lack of ablation experiments on potential gradient conflicts or modality competition caused by expanding the vocabulary.**
>
> **R2:** We sincerely thank the reviewer for raising this insightful point. The primary motivation for expanding the vocabulary is to achieve a **truly unified, end-to-end** multimodal system. Without a unified vocabulary, we would have to rely on modality-specific external generation decoders, which would compromise the end-to-end nature of our model.
>
> We fully understand the reviewer’s concern. Following the suggestion, **we conducted an ablation study to investigate the existence of modality competition in our model.** Specifically, we start from an intermediate checkpoint and continue training under two settings:
>
> - **Baseline (Ours):** Normal joint training with all gradients computed.
> - **Ours w/o Visual-Text Gradients:** We zeroed out the gradients from visual-text tasks, updating the model only on speech-text tasks.
>
> The model’s performance after training for 500 steps is shown below:
>
> |  | TTS WER $\downarrow$ | ASR WER $\downarrow$ |
> | --- | --- | --- |
> | Baseline | 3.90 | 7.10 |
> | Ours w/o Visual-Text Gradients | 3.77 | 7.06 |
>
> The marginal performance gap under two different settings indicates that modality conflicts are **not significant** in our model. We attribute this to our **Three-Stage Progressive Training Pipeline**, which introduces modalities progressively rather than mixing them simultaneously from scratch. In future work, we will explore more advanced architecture designs and training strategies to further alleviate this issue.
>
> **Q3:  What are the specific advantages and disadvantages of Any-Diffusion regarding computational costs (training memory, inference KV-cache) compared to AR models?**
>
> **R3:** We thank the reviewer for raising this insightful question. We provide a detailed analysis of the advantages and disadvantages of diffusion models compared to AR models regarding **inference and training costs** below:
>
> 1. **Inference Cost**
>     - **Advantages: Parallel Decoding Capability.** AR models decode one token per forward pass. In contrast, diffusion models are able to decode multiple tokens in parallel, thereby **achieving low latency by setting a small number of inference steps.** We present the latency of our model compared with AR baselines on the text-to-image generation task below. When reducing inference steps, our method achieves **significantly lower latency** compared with AR baselines, with **minor degradation in performance.**
>
>
>         | Method | #Param | Average Latency $\downarrow$ | CLIP-T $\uparrow$ |
>         | --- | --- | --- | --- |
>         | Emu | 14B | 3.022 | 0.286 |
>         | NeXT-GPT | 7B | 6.227 | 0.225 |
>         | Ours (256 steps) | 7B | 28.57 | 0.236 |
>         | Ours (10 steps) | 7B | 1.29 | 0.233 |
>     - **Disadvantages: Absence of KV Cache.** Standard diffusion models lack a KV-Cache mechanism, requiring more FLOPs per forward pass than AR models. However, recent pioneering work [1] has introduced KV-cache equivalents designed for discrete diffusion models. We will explore integrating the advanced inference algorithm to further optimize inference efficiency in future work.
> 2. **Training Cost**
>
>     In terms of training memory usage, Any-Diffusion and AR models are actually **identical**. During training, the GPU memory is determined by the model size, batch size, and sequence length. Since both Any-Diffusion and AR models utilize standard transformer backbones, the training memory usage of these two paradigms is the same.
>
>
> [1]. Fast-dllm: Training-free acceleration of diffusion llm by enabling kv cache and parallel decoding.

---

> > ### Author Rebuttal · Reviewer_qyGk · 2026-04-04
> >
> > please see my new comment

---

> > > ### Author Response · Authors · 2026-04-08
> > >
> > > We thank you for the constructive comments and are glad to see that our responses have resolved the raised concerns.

---

### Official Review · Reviewer_SMPX · 2026-03-12

**Soundness:** 3
**Presentation:** 3
**Significance:** 3
**Originality:** 2
**Overall Recommendation:** 4
**Confidence:** 5

**Summary:**

This paper introduces Any-Diffusion, an any-to-any multimodal model that replaces traditional autoregressive architectures with a mask-based discrete diffusion model to achieve unified comprehension and generation across text, image, and speech. By modeling the joint distribution of multimodal discrete tokens in a shared semantic space, the framework eliminates the need for auxiliary output models, enabling intrinsic alignment and flexible generation capabilities. To support this architecture, the authors develop specialized training and inference techniques—including a progressive training pipeline and modality-specific strategies like position penalties and token pre-infilling, demonstrating that diffusion-based models can rival or exceed the performance of state-of-the-art autoregressive multimodal systems.

**Compliance With Llm Reviewing Policy:**

Affirmed.

**Final Justification:**

Overall, the paper would need additional clarifications, inclusion of additional experiments and rewrites in the paper. Thus, I maintain my score.

**Key Questions For Authors:**

Refer to the weaknesses

**Limitations:**

While the paper lacks exhaustive benchmarking and theoretical depth, I recommend its acceptance as a pioneering exploration of discrete diffusion for any-to-any multimodal modeling. This work provides a valuable alternative to the dominant autoregressive paradigm, and its encouragement of diverse probabilistic modeling approaches will significantly benefit the research community.

**Strengths And Weaknesses:**

Strengths:
1. Architectural Innovation: The paper introduces a pioneering any-to-any multimodal framework that replaces traditional autoregressive architectures with a unified mask-based discrete diffusion model, enabling intrinsic semantic alignment across text, image, and speech.
2. Specialized Optimization: The authors develop a suite of tailored training and inference techniques such as the three-stage progressive pipeline, position penalties, and token pre-infilling that effectively overcome the unique challenges of discrete diffusion for multimodal generation.
3.Empirical Validation: Through comprehensive experiments, the work demonstrates that Any-Diffusion achieves performance competitive with some classical autoregressive systems, providing a significant proof-of-concept for the viability of diffusion-based multimodal intelligence.

Weaknesses:
1. While Any-Diffusion is presented as a pioneering any-to-any discrete diffusion model, the experimental evaluation is insufficient; it lacks head-to-head comparisons with the latest state-of-the-art benchmarks and models specifically designed for image-text or text-speech tasks, which are necessary to establish its competitive standing
2. The paper fails to provide a transparent comparison of inference latency against autoregressive baselines, particularly in the absence of specialized engineering optimizations (e.g., KV-cache equivalents for diffusion). Given that mask-based diffusion typically requires iterative decoding, it is likely slower than autoregressive models in practice, and the authors should provide a rigorous analysis of this speed-accuracy trade-off.
3. The distinction between the proposed mask-based diffusion and existing flow-based or alternative diffusion frameworks remains theoretically thin, as these approaches are often mathematically equivalent (representing different noise-modeling paradigms) [1]. The authors need to provide a deeper classification and justification for why this specific mask-based formulation offers unique advantages over contemporary unified frameworks like NExT-OMNI, rather than claiming novelty based on superficial architectural differences.


[1] Diffusion Models and Gaussian Flow Matching: Two Sides of the Same Coin

---

> ### Author Rebuttal · Authors · 2026-03-31
>
> We appreciate the reviewer's positive feedback on our work. Our point-by-point responses to all comments are provided below. We will add the discussion and additional results in the final version.
>
> **Q1: Lack of comparisons with the latest models.**
>
> **R1:** We appreciate the valuable comment and agree that comparison with SOTA models strengthens our contribution. While our model demonstrates strong performance among various any-to-any LLMs, we acknowledge a performance gap compared to advanced enterprise models like the Qwen series. However, we would like to highlight our model's **superiority in the following aspects** despite this gap:
>
> 1. **Strong comprehensive performance**
>
>     Any-Diffusion outperforms **specialized speech LLMs** (GLM-4-Voice) on the TTS task. For image-text tasks, it matches widely recognized **visual LLMs** (e.g., LLaVA, InstructBLIP, Emu) and outperforms existing **any-to-any models** (e.g., AnyGPT, NExT-GPT) on VQA.
>
> 2. **Remarkable Data Efficiency**
>
>     Any-Diffusion achieves comprehensive capabilities using only **75.18B** training tokens, which is significantly fewer than other baselines (**GLM-4-Voice: 1055B; Emu: 150B; AnyGPT: 176.4B**). Please see **R3 to Reviewer ukSj** for details.
>
>
> In summary, our motivation is to pioneer a **unified, any-to-any multimodal framework built purely on discrete diffusion.** Our model achieves **comprehensive performance with extremely limited computational resources**, highlighting the potential of discrete diffusion for multimodal systems.
>
> **Q2: Lack of latency comparison against AR baselines and speed-accuracy trade-off analysis.**
>
> **R2:**
>
> 1. **Speed-performance trade-off**
>
> We sincerely appreciate the constructive suggestion. We extend the results from Table 3 to present the **speed-performance trade-off** as follows:
>
> Text to Image Generation (COCO):
>
> | **Steps** | **Avg Latency (s)** $\downarrow$ | **CLIP-T / I** $\uparrow$ |
> | --- | --- | --- |
> | 256 | 28.57 | 0.2360 / 0.6614 |
> | 50 | 5.52 | 0.2354 / 0.6593 |
> | 10 | 1.29 | 0.2325 / 0.6524 |
>
> Speech to Image Generation (COCO):
>
> | **Steps** | **Avg Latency (s)** $\downarrow$ | **CLIP-T / I** $\uparrow$ |
> | --- | --- | --- |
> | 256 | 39.90 | 0.2322 / 0.6450 |
> | 50 | 7.74 | 0.2323 / 0.6454 |
> | 10 | 4.25 | 0.2289 / 0.6376 |
>
> Text-to-Speech (LibriTTS):
>
> | **Steps** | **Average RTF** $\downarrow$ | **WER** $\downarrow$ |
> | --- | --- | --- |
> | 0.5 $L$ | 0.4265 | 2.22 |
> | 0.25 $L$ | 0.3580 | 2.67 |
> | 0.125 $L$ | 0.3407 | 5.54 |
>
> where $L$ denotes the sequence length, and RTF is estimated by $\text{RTF}=\frac{\text{Inference Latency}}{\text{Generated Speech Duration}}$.
>
> 2. **Latency comparison with AR baselines**
>
> Based on the results above, **we compared the inference latency of our method against AR models across different modalities.** For the TTS task, the RTF of GLM-4-Voice (9B) is 0.6855. For the text-to-image task, the latencies of Emu (14B) and NeXT-GPT (7B) are 3.022s and 6.227s. As shown in the above table, our model demonstrates **significantly lower latency than AR baselines** under few-step settings.
>
> 3. **Discussion on the absence of engineering optimizations**
>
> We fully agree with the reviewer’s comment that our current implementation lacks optimizations like KV-cache equivalents for diffusion. Nevertheless, **our method achieves the superior generation speeds shown above relying purely on the native parallel decoding mechanism, without additional engineering techniques.** We will explore advanced diffusion acceleration algorithms to further improve our model in the future.
>
> **Q3: Justification for the unique advantages of the mask-based formulation over discrete flow matching (DFM).**
>
> **R3:** We appreciate the insightful comment. We agree that mask-based diffusion and DFM are mathematically related. However, we clarify that the **mask-based formulation exhibits different inference dynamics, offering unique advantages for multimodal foundation models.** Specifically, our model utilizes mask token prediction for training, enabling the model to condition on partially known tokens during inference. This inherent characteristic allows us to introduce **special token pre-infilling** to improve the generation quality in spoken tasks. This characteristic also enables our model to perform visual downstream tasks, such as inpainting. In contrast, DFM is optimized to continuously update the entire sequence of tokens at every step along the probability flow. **It is mathematically non-trivial for DFM to predict based on partially known tokens**, as it would lead to severe train-test distribution inconsistency.
>
> **Q4: Lack of discussion on limitations.**
>
> **R4:** We sincerely thank the reviewer for the positive recognition of our work and this valuable comment. We will include the above discussion into our paper to improve the theoretical depth. We acknowledge that including more baselines would strengthen the contribution, and will include this limitation into paper.

---

> > ### Author Rebuttal · Reviewer_SMPX · 2026-04-03
> >
> > the authors  still need to explicitly address two points:
> >
> > **Compute–quality trade-off** Any-Diffusion’s strong results (even without KV-cache) rely on a more capable backbone model. It is theoretically unrealistic for a weaker base model, lacking such optimizations, to match this performance under equal or lower compute budgets. Please emphasize and quantify this trade-off honestly.
> >
> > **DFM vs. mask-based diffusion** Although both inject discrete noise in a mathematically equivalent way, their inference dynamics differ. Mask-based diffusion naturally supports partial-token conditioning (pre-infilling, inpainting) without train–test mismatch, whereas DFM’s full-sequence updates cannot easily “freeze” known tokens. A concise discussion of these distinctions and their respective use-case scopes is needed.
> >
> > Overall, the paper would need additional clarifications, inclusion of additional experiments and rewrites in the paper. Thus, I maintain my score.

---

> > > ### Author Response · Authors · 2026-04-08
> > >
> > > We sincerely thank you for the follow-up comments. Below we address the new concerns.
> > >
> > > **Q1: Compute–quality trade-off**
> > >
> > > **R1:** Thanks for this insightful comment. While we agree that a strong backbone is beneficial for improving the model’s performance, we would like to clarify that the backbone does not have multimodal capabilities, and our model’s performance comes from various aspects beyond the backbone. A core contribution of our work is **validating the effectiveness of discrete diffusion models under limited compute budgets.** Specifically, the model's performance comes from two main aspects:
> > >
> > > - In terms of architecture, the **inherent random mask mechanism** of discrete diffusion **relaxes the causal inductive bias of AR models**, increasing the freedom to fit the training data. This is a fundamental advantage of discrete diffusion models, as discussed in [1].
> > > - In terms of optimization, we introduce several **tailored training and inference strategies**. Our partial ablation studies already demonstrate their impact: removing the **position penalty** drops text-to-image CLIP-I/T scores from 0.236/0.662 to 0.222/0.647, and removing **special-token pre-infilling** causes the model to generate overly brief responses (e.g., the first case in Figure 4 degrades to merely "*The woman is wearing jeans*").
> > >
> > > Finally, **to quantify the backbone's impact, we are currently running ablation studies using alternative backbones**, including the diffusion backbone LLaDA and AR backbones like Vicuna and Qwen. Given the substantial computational resources and time required, these experiments cannot be completed within the short rebuttal period. We plan to include these results in the final version of the paper.
> > >
> > > [1]. Ni, Jinjie, et al. "Diffusion language models are super data learners." *arXiv preprint arXiv:2511.03276* (2025).
> > >
> > > **Q2: DFM vs. mask-based diffusion**
> > >
> > > **R2:** We sincerely thank you for this constructive suggestion. We clarify the distinctions and use-case scopes of mask-based diffusion and DFM as follows:
> > >
> > > - **Inference distinctions between mask-based diffusion models and DFM**. While both frameworks model the transition from a noise distribution to a target data distribution, their decoding processes differ fundamentally. **Mask-based diffusion models utilize a mask token prediction objective** for training. During inference, they **iteratively decode and freeze the Top-k highest-confidence tokens** at each step. The newly decoded tokens immediately serve as clean, reliable context for the remaining masked tokens. In contrast, DFM operates via **full-sequence continuous updates**. During training, DFM optimizes the probability velocity for the entire sequence, while mask-based diffusion models compute the loss only on masked tokens. During inference, DFM refines the **entire sequence** simultaneously at every step without freezing individual tokens. Consequently, forcing known tokens into the DFM sampling process would introduce train-test mismatch, as it is trained by predicting the logits of the entire token sequence.
> > > - **Respective use-case scopes.** The decoding mechanism of mask-based diffusion models makes them inherently suited for **constrained generation conditioned on partially known tokens**, such as visual inpainting and template-based generation. In contrast, DFM adopts an iterative global refinement process that updates the entire sequence at every step. This mechanism enables the model to refine the tokens throughout the entire generation process, which closely aligns with continuous diffusion models [1]. Consequently, **DFM excels in tasks where continuous diffusion models traditionally demonstrate strong performance**, such as visual context generation (e.g., text-to-video or text-to-image).
> > >
> > > We hope that the added discussion will address your concerns. We will include the comparison between the DFM and mask-based diffusion in the related work of our final version.
> > >
> > > [1]. Deng, Haoge, et al. "Uniform discrete diffusion with metric path for video generation." *arXiv preprint arXiv:2510.24717* (2025).

---

### Official Review · Reviewer_ukSj · 2026-03-13

**Soundness:** 3
**Presentation:** 3
**Significance:** 3
**Originality:** 3
**Overall Recommendation:** 4
**Confidence:** 4

**Summary:**

This paper presents Any-Diffusion, a masked discrete diffusion based any-to-any multimodal model that aims to unify understanding and generation across text, image, and speech. The key idea is to convert all three modalities into discrete tokens and directly model their joint distribution, instead of using the common architecture of an LLM plus extra modality-specific decoders. The paper also introduces a three-stage training pipeline, the SDVI dataset, attenuated tail-pad masking, and several inference techniques to improve image and speech generation quality and sampling efficiency. The authors claim competitive or better results on ASR, TTS, VQA, text-to-image, and speech-to-image tasks.

**Compliance With Llm Reviewing Policy:**

Affirmed.

**Key Questions For Authors:**

“In the text-to-image task, our method achieves superior text-image alignments compare to other any-to-any models, and visual quality comparable to methods that rely on external pretrained diffusion models.” However “visual quality comparable ” is not discussed in the paper, e.g. by human evaluation or other evaluation reported.

“We examine the model’s ability to achieve unified alignment across the speech, image, and text modalities by evaluating its performance on speech-to-image generation.” then “As shown in Table 3, Any-Diffusion achieves similar generation quality condition on text and speech…” However, the results is on  text-conditioned and speech-conditioned on the proposed model, not mentioned other public benchmarks.

**Limitations:**

Limitations is not discussed. It would be more appealing to acknowledge the limitations of the proposed method, e.g., compared to state-of-the-art autoregressive models or transfusion models.

**Strengths And Weaknesses:**

# Strengths
* The problem setting is meaningful. Compared to many other unified multimodal systems that rely on autoregressive LLMs, this work explicitly explores a different architectural route by using discrete diffusion to directly model the joint distribution of multimodal discrete tokens. This direction has some novelty.
* The paper does not only study text-image or speech-text tasks, but attempts to support ASR, TTS, VQA, text-to-image, speech-to-image, and spoken visual interaction in a single framework. The construction of the SDVI data for spoken visual interaction also makes the system more complete than standard multimodel evaluations.
* The reported results show competitive performance across several benchmarks, and the sampling-step study also supports the claimed efficiency potential of diffusion-based generation.

# Weaknesses
My main concern is that the methodological novelty and the experimental support are not fully matched.
1. Although the paper introduces many design choices, including the three-stage training scheme, SDVI, attenuated tail-pad masking, position penalty, special token pre-infilling, and adaptive token length assignment, it does not provide systematic ablations to show the contribution of each component.
2. There is still clear unclearness in evaluation. Visual generation is mainly assessed with CLIP-based metrics, without stronger generation evaluation or human studies. Speech-to-image and spoken visual interaction are supported mostly by qualitative results and limited automatic metrics, without sufficiently direct and strong comparisons. For a paper whose main claim is unified any-to-any multimodal modeling, these cross-modal settings should be among the strongest pieces of evidence.
3. Some comparisons are not fully convincing in terms of fairness. Several baselines differ in scale, data, or task coverage, and the paper does not provide enough detail on training mixture, total token scale, or training cost. As a result, it is difficult to judge how much of the gain comes from the architecture itself versus data scale and engineering choices.
Finally, the writing needs improvement. There are noticeable grammar and phrasing issues, and some methodological explanations remain heuristic rather than rigorous, which weakens the overall clarity and persuasiveness.

---

> ### Author Rebuttal · Authors · 2026-03-31
>
> We sincerely thank the reviewer for the positive recognition of our work. Our point-by-point responses are provided below. We will include the discussion in our final version.
>
> **Q1: Lack of ablation studies.**
>
> **R1:** We provide the ablations for several key designs below.
>
> - **Position Penalty:** We conducted an ablation study of position penalty on the text-to-image task. Without it, the CLIP-T / I scores reduce from **0.236 / 0.662** to **0.222 / 0.647**, which confirms its effectiveness in enhancing visual generation quality.
> - **Special Token Pre-infilling:** We provide qualitative ablations on special token pre-infilling in the spoken VQA task based on **the first example in Figure 4**. Without special token pre-infilling, the transcription of the speech response becomes overly brief (e.g., "*The woman is wearing jeans*"). We will provide more cases in the revised paper.
>
> **Discussion on Adaptive Token Length Assignment:** We clarify that this strategy is essential for ASR and TTS. These tasks have highly variable data lengths. A fixed token length forces excessive padding or truncates information, which is a known issue in discrete diffusion [1].
>
> **Discussion on the training strategies:** We agree that training strategies ablation would be valuable.  However, conducting full ablations **is difficult to complete within rebuttal due to the extreme compute demand**. We plan to include them in the final version.
>
> [1]. Dimple: Discrete diffusion multimodal large language model with parallel decoding
>
> **Q2: Insufficient evaluation for visual generation and cross-modal tasks.**
>
> **R2:** We appreciate the constructive comment. We conducted **a human study on image generation**, and **a quantitative evaluation on the spoken VQA task**, as detailed below.
>
> 1. **Human study on image generation**
>
>     We invited several participants to rate the **Prompt Alignment and Aesthetic Quality** (1-5 scale) of images. Our model achieves **comparable aesthetic quality and better prompt alignment** than baselines. In addition, it **demonstrates strong generation performance conditioned on both text and speech**. We will expand this evaluation in future work to include more participants and baselines.
>
>     | Method | Prompt Alignment | Aesthetic Quality |
>     | --- | --- | --- |
>     | Emu | 3.66 | 3.97 |
>     | NeXT-GPT | 3.63 | 3.96 |
>     | Ours (text-to-image) | 3.83 | 3.89 |
>     | Ours (speech-to-image) | 3.89 | 3.91 |
> 2. **Quantitative evaluation on Spoken VQA**
>
>     Since there is no benchmark for Spoken VQA, we converted MME text questions into speech for evaluation. The MME-P of our model on VQA and Spoken VQA are **1176.1** and **967.1** respectively. These results demonstrate that **our method maintains strong spoken visual interaction capability.**
>
>
> **Q3: Discussion on comparison experiments, training details, source of gains, and writing issues.**
>
> **R3:** We appreciate the valuable feedback and address these concerns below:
>
> 1. **Training details**
>
>     During training, we set the batch size and max length to **128 and 3K**, respectively. The training steps for the three stages are **27.6K, 54.2K, and 114K**. Based on this configuration, our model uses **75.18B** training tokens across three stages.
>
> 2. **Comparison on training cost and task coverage.**
>
>     Our **75.18B** training tokens are significantly fewer than baselines (**GLM-4-Voice: 1055B; Emu: 150B; AnyGPT: 176.4B**). Please note that NeXT-GPT, LLaVA, and InstructBLIP are excluded here as their training tokens are not disclosed. Any-Diffusion covers **a broader range of tasks than specialized Speech LLMs or Visual LLMs**. Although some advanced baselines support video understanding that is not included in our model, our method **still exhibits strong superiority by achieving comprehensive capabilities under very limited compute constraints**.
>
> 3. **Disentangling the gains.**
> Our model uses fewer training tokens than baselines, thus **the performance gains do not stem from data scaling**. Instead, the advantages originate from the **mask-based discrete diffusion architecture and our tailored strategies**. For instance, the bidirectional attention inherently benefits global context comprehension, while the optimization strategies address the unique behaviors of discrete diffusion.
> 4. **Rigor and Writing**
>
>     We appreciate the reviewer's careful reading. We will correct grammatical errors and improve rigor in the revised version.
>
>
> **Q4 & Q5. Visual quality metrics and speech-to-image benchmarks.**
>
> **R4 & R5:** Due to **the lack of benchmarks on speech-to-image,** we conducted a **human study** (please see **R2**) demonstrating comparable visual quality to AR baselines and strong speech-to-image performance.
>
> **Q6: Missing discussion on limitations**
>
> **R6:** We appreciate this suggestion. We will add a limitations section discussing the need for AR and transfusion baselines comparisons.

---

> > ### Author Rebuttal · Reviewer_ukSj · 2026-04-04
> >
> > Thanks for the detailed response and additional evaluation results. I have no further questions.

---

> > > ### Author Response · Authors · 2026-04-08
> > >
> > > We sincerely thank you for the thoughtful review and are pleased that our responses have addressed the concerns raised.

---

### Decision · Program_Chairs · 2026-04-30

**Decision:**

Accept (regular)

**Comment:**

The paper proposes Any-Diffusion, which is a unified any-to-any multimodal model built on mask-based discrete diffusion, aiming to move beyond conventional autoregressive architectures. By modeling the joint distribution over discrete tokens, the framework naturally supports flexible multimodal understanding and generation across text, speech, and images, including multimodal compositions beyond bimodal settings. Empirical results across diverse benchmarks show competitive or superior performance of Any-Diffusion.

This paper provides sufficient insights into the research field of multimodal foundation models, and its engineering contribution is considerable. The reviewers reached a consensus and accepted the work. The AC checked it and agreed with the reviewers. Please remember to incorporate the additional experiments and results from the rebuttal into the final version of the work.